# Prevalence and Characteristics of Spinal Sagittal Malalignment in Patients with Osteoporosis

**DOI:** 10.3390/jcm10132827

**Published:** 2021-06-26

**Authors:** Takayuki Matsunaga, Masayuki Miyagi, Toshiyuki Nakazawa, Kosuke Murata, Ayumu Kawakubo, Hisako Fujimaki, Tomohisa Koyama, Akiyoshi Kuroda, Yuji Yokozeki, Yusuke Mimura, Eiki Shirasawa, Wataru Saito, Takayuki Imura, Kentaro Uchida, Yuta Nanri, Kazuhide Inage, Tsutomu Akazawa, Seiji Ohtori, Masashi Takaso, Gen Inoue

**Affiliations:** 1Department of Orthopaedic Surgery, School of Medicine, Kitasato University, Kanagawa 252-0374, Japan; auaa.0718.aaui@gmail.com (T.M.); nakazawa@kitasato-u.ac.jp (T.N.); nineball5121@yahoo.co.jp (K.M.); ayumukawakubo0827@gmail.com (A.K.); hisako19830608@yahoo.co.jp (H.F.); tomohisakoyama1989@gmail.com (T.K.); akiyoshikvroda@yahoo.co.jp (A.K.); yuji0328yoko@yahoo.co.jp (Y.Y.); msm.men.36@gmail.com (Y.M.); eeiikkii922@yahoo.co.jp (E.S.); boatwataru0712@gmail.com (W.S.); tk2003@kitasato-u.ac.jp (T.I.); kuchida@med.kitasato-u.ac.jp (K.U.); mtakaso@kitasato-u.ac.jp (M.T.); ginoue@kitasato-u.ac.jp (G.I.); 2Department of Rehabilitation, Kitasato University Hospital, Sagamihara 252-0374, Japan; nanriyuta.rpt-103@hotmail.co.jp; 3Department of Orthopedic Surgery, Graduate School of Medicine, Chiba University, Chiba 260-8670, Japan; kazuhideinage@chiba-u.jp; 4Department of Orthopaedic Surgery, School of Medicine, St. Marianna University, Kawasaki 216-8511, Japan; cds00350@par.odn.ne.jp (T.A.); sohtori@faculty.chiba-u.jp (S.O.)

**Keywords:** spinal sagittal alignment, osteoporosis, low back pain, health-related quality of life

## Abstract

Spinal sagittal malalignment due to vertebral fractures (VFs) induces low back pain (LBP) in patients with osteoporosis. This study aimed to elucidate spinal sagittal malalignment prevalence based on VF number and patient characteristics in individuals with osteoporosis and spinal sagittal malalignment. Spinal sagittal alignment, and VF number were measured in 259 patients with osteoporosis. Spinal sagittal malalignment was defined according to the SRS-Schwab classification of adult spinal deformity. Spinal sagittal malalignment prevalence was evaluated based on VF number. In patients without VFs, bone mineral density, bone turnover markers, LBP scores and health-related quality of life (HRQoL) scores of normal and sagittal malalignment groups were compared. In 205 of the 259 (79.2%) patients, spinal sagittal malalignment was detected. Sagittal malalignment prevalence in patients with 0, 1, or ≥2 VFs was 72.1%, 86.0%, and 86.3%, respectively. All LBP scores and some subscale of HRQoL scores in patients without VFs were significantly worse for the sagittal malalignment group than the normal alignment group (*p* < 0.05). The majority of patients with osteoporosis had spinal sagittal malalignment, including ≥70% of patients without VFs. Patients with spinal sagittal malalignment reported worse LBP and HRQoL. These findings suggest that spinal sagittal malalignment is a risk factor for LBP and poor HRQoL in patients with osteoporosis.

## 1. Introduction

Patients with osteoporosis often report low back pain (LBP), particularly intermittent LBP such as vague LBP due to standing or walking for a long stretch of time. In clinical settings, the types of LBP reported tend be difficult to treat. Whether osteoporosis causes LBP is controversial because its pathological mechanism has not been fully elucidated. Several factors, including high bone turnover [1], low muscle mass [2], and vertebral fractures (VFs) [3], have been reported to be associated with increased risk of LBP and osteoporosis. In addition, it is well known that VFs induce spinal sagittal malalignment in osteoporosis patients [4]. 

Patients with spinal sagittal malalignment presenting as adult spinal deformity often complained of severe LBP that induced the deterioration of health-related quality of life (HRQoL), requiring treatment [5]. Patients with spinal deformity tend to be elderly individuals who also have osteoporosis. Therefore, osteoporosis might be associated with adult spinal deformity. Furthermore, it has been reported that osteoporotic patients with VFs showed worse spinal sagittal alignment and LBP and HRQoL scores [6]. However, the prevalence of spinal sagittal malalignment in osteoporosis patients remains unclear. Improving our understanding of the characteristics of patients with spinal sagittal malalignment may lead to the improvement of spinal sagittal malalignment treatment. The aim of this study was to elucidate the prevalence of spinal sagittal malalignment based on VF number and the characteristics of patients with osteoporosis and spinal sagittal malalignment. 

## 2. Materials and Methods

### 2.1. Patient Population

The records of patients with osteoporosis who first visited these facilities from June 2015 to March 2017 were reviewed in this cross-sectional study. We excluded patients who developed new vertebral fractures within three months. The remaining 259 patients (48 men, 211 women; mean age: 71.5 years) were included.

### 2.2. Measurements

In all patients, we evaluated bone mineral density (BMD) of the lumbar spine (LS: L2–L4), femoral neck (FN), and total hip (TH), using dual-energy X-ray absorptiometry (DXA: Horizon DXA System; Hologic Inc., Santa Clara, CA, USA), and serum levels of bone turnover markers including bone-specific alkaline phosphatase (BAP; Beckman Coulter Inc. Brea, CA, USA) and tartrate-resistant acid phosphatase 5b (TRACP5b; DS Pharma Biomedical Inc., Osaka, Japan). 

#### 2.2.1. Radiographical Evaluation

X-ray images taken in frontal and lateral views of the whole spine, including the hip joints, in the standing position were reviewed to evaluate spinal sagittal alignment and VFs. For the evaluation of spinal sagittal alignment, three spinal sagittal alignment parameters were measured. To assess pelvic tilt (PT), the angle between the line joining the midpoint of the bilateral center of the femoral head to the center of the S1 endplate and a vertical reference line was measured. Pelvic incidence (PI) was determined by measuring the angle between a line joining the midpoint of the bilateral center of the femoral head to the center of the S1 endplate, and a line orthogonal to the S1 endplate, as previously reported [7]. To measure lumbar lordosis (LL), we assessed the angles between the first line parallel to the upper endplate of L1 and the second line parallel to the superior endplate of the sacral base on lateral views of the whole-spine radiograph. Then, PI-LL was used to evaluate spinal sagittal alignment. For measuring the sagittal vertical axis (SVA), the horizontal distance between the posterior-superior corner of the sacrum and a vertical line from the center of C7 was measured, as previously reported [8]. In accordance with the SRS-Schwab classification scheme [9], patients were categorized using three sagittal spinopelvic modifiers, including PT, PI-LL, and SVA. SVA > 40 mm, PT > 20°, or PI-LL > 10° was defined as spinal sagittal malalignment. Based on these data, subjects were divided into a normal alignment group and a sagittal malalignment group.

#### 2.2.2. Clinical Outcome Evaluation

LBP was evaluated using the Japanese Orthopedic Association Back Pain Evaluation Questionnaire (JOABPEQ), the Oswestry Disability Index (ODI), and the visual analogue scale (VAS). JOABPEQ consists of five functional scores: pain-related disorders, lumbar spine dysfunction, gait disturbance, social life dysfunction, and psychological disorders. Each domain score ranges from 0 to 100, and higher scores corresponded to an improved patient condition.

In addition, HRQoL was evaluated using the MOS 36-Item Short-Form Health Survey (SF-36). SF-36 consists of 8 subscales, as follows: physical function, PF; role physical, RP; body pain, BP; general health, GH; vitality, VT; social functioning, SF; role emotional, RE; and mental health, MH. The score for each domain ranged from 0 to 100, and higher scores indicated a better condition.

### 2.3. Statistical Analysis

First, the prevalence of a spinal sagittal malalignment was evaluated based on the number of VFs identified. Factors including age, BMD, serum levels of bone turnover markers, and parameters of spinal sagittal alignment of the three groups of patients with 0, 1, and ≥2 VFs were compared using the Bonferroni test for multiple comparisons. Sex and the spinal sagittal alignment differences were compared using the chi-squared test.

Characteristics of spinal sagittal malalignment evaluated in sagittal malalignment and normal alignment groups in patients without VFs were compared. Leven’s test was used to assess variance for variables of interest. To assess data with unequal variance, the Mann–Whitney U test was applied. An unpaired *t*-test was used to assess data with equal variance. All data were analyzed using IBM SPSS Statistics version 26 (IBM, Atmonk, NY, USA), and *p* < 0.05 were considered significant.

### 2.4. Ethics

Ethical approval from Institutional Review Board in Kitasato University was obtained for this study (Approval code, #B17–197), which was conducted in accordance with the ethical principles specified in the 1964 Declaration of Helsinki and its later amendments.

## 3. Results

Characteristics of the patient population and BMD, serum levels of bone turnover markers, and parameters of spinal sagittal alignment, are listed in Table 1. Spinal malalignment was observed in 205 of 259 (79.2%) patients. BMDs of the FN and TH in the group with ≥2 VF were significantly lower than those of the 0 group. The BMD of the TH in the group with 1 VF was significantly lower than that of the 0 VF group (*p* < 0.05).

With regard to spinal sagittal alignment parameters, PT and SVA values of the ≥2 VF group was significantly higher than that of the 0 VF group (*p* < 0.05). No significant differences between 0, 1, ≥2 VF groups were observed with regard to BMD, bone turnover markers, and PI-LL (*p* > 0.05). The prevalence of a spinal sagittal malalignment in patients with 0, 1, or ≥2 VFs was 72.1%, 86.0%, and 86.3%, respectively, and differences among the three groups were determined as significant (*p* < 0.05) (Table 1).

With regard to LBP and the HRQoL score, all five domains of JOABPEQ, ODI, and VAS, as well as all eight subscales of the SF-36 of the ≥2 VF group, were significantly worse than those of the 0 VF group (*p* < 0.05). In addition, the values of all five domains of JOABPEQ, ODI, RP, BP, SF, RE, and MH of SF-36 in the 1 VF group were significantly worse than those in the 0 VF group (*p* < 0.05) (Table 1).

In a sub-analysis of patients without VFs, no significant differences were observed between the sagittal malalignment group and the normal alignment group with regard to age; LS, FN, and TH of BMD; or bone turnover markers, including BAP and TRACP5b (*p* > 0.05) (Figure 1). In contrast, all five JOABPEQ functional scores (including pain-related disorders, lumbar spine dysfunction, gait disturbance, social life dysfunction, and psychological disorders) of the sagittal malalignment group were significantly lower than those of the normal alignment group (*p* < 0.05) (Figure 2A). Furthermore, ODI and the VAS values determined for LBP in patients without VFs were significantly higher in the sagittal malalignment group than in the normal alignment group (*p* < 0.05) (Figure 2B,C). Additionally, PF, RP, VT, RE, and MH of SF-36 values of the sagittal malalignment group were significantly lower than those of the normal alignment group (*p* < 0.05) (Figure 3).

## 4. Discussion

In the current study, the prevalence of spinal sagittal malalignment in osteoporosis patients was determined to be 79.2%. Furthermore, as the number of VFs increased, the prevalence of spinal sagittal malalignment also increased and LBP and HRQoL scores worsened. Interestingly, more than 70% of osteoporosis patients without VFs had spinal sagittal malalignment. Additionally, in patients without VFs, patients with spinal sagittal malalignment had worse LBP and HRQoL scores than patients with normal alignment.

Regarding the relationship between VFs and spinal sagittal alignment, as the number of VFs increased, the prevalence of spinal sagittal malalignment also increased, and LBP and HRQoL scores worsened in this study. Mochizuki et al. previously reported that spinal sagittal alignment is associated with age and VF in patients with rheumatoid arthritis [10]. In addition, osteoporosis patients with VFs have worse global sagittal alignment and a worsened quality of life [6]. Scaturro et al. reported that the severity of LBP is correlated with the number of vertebral fractures [11]. These findings indicate that VFs are closely correlated with sagittal spinal malalignment and affect LBP as well as HRQoL. 

With regard to cause-and-effect relationships between VFs and spinal sagittal malalignment, Zhang et al. reported that multiple VFs lead to spinal sagittal malalignment in patients with osteoporosis [12]. In contrast, several authors reported that spinal sagittal malalignment was a potential risk factor for increased VF incidence in patients with osteoporosis [4,13,14]. These findings indicate that VFs induce spinal sagittal malalignment; spinal sagittal malalignment also leads to VFs in patients with osteoporosis. 

In the current study, more than 70% of patients with osteoporosis without VFs had spinal sagittal malalignment. In a longitudinal study with a minimum of 10 years of follow-up, Takeda et al. reported that spinal sagittal malalignment, decreased lumbar lordosis, and increased SVA were correlated with age in patients without VFs [15]. Regarding the underlying mechanism of spinal sagittal malalignment in patients without VFs, several authors reported a relationship between spinal sagittal malalignment and decreased muscle mass in patients with spinal diseases [16,17]. Additionally, Scaturro et al. reported that combination treatments with medication and postural training/resistance exercises showed improvements in the pain and QoL for patients with osteoporosis undergoing rehabilitation [18]. These findings indicate that decreasing muscle mass may induce spinal sagittal malalignment.

In the current study, the majority of osteoporosis patients had spinal sagittal malalignment. In recent years, several authors reported that long spinal fusion and corrective surgery for spinal sagittal malalignment could be used to achieve good spinal alignment. Improvements were due to recent, remarkable developments in surgical techniques and spinal instruments and contributed to improvements in ADL and LBP outcomes [19,20]. However, high perioperative complication rates for long spinal fusion and corrective surgery have been reported [21]. Therefore, performing the highly invasive and costly surgery in all osteoporosis patients is not advisable. Alternatively, we should consider early intervention for spinal sagittal malalignment in osteoporosis patients, which may prevent the need for surgery to correct adult spinal deformity. 

When investigating relationships between spinal sagittal alignment and LBP or HRQoL, Schwab et al. reported that high SVA, PI-LL, and PT values induced the deterioration of HRQoL in elderly adult patients with spinal deformity and a defined SRS-Schwab classification [9]. Similarly, the current study reported that osteoporosis patients with spinal sagittal malalignment and a defined SRS-Schwab Classification had some reduced HRQoL subscale values, including PF, RP, VT, RE, and MH. On the other hand, results of a meta-analysis by Chun et al. indicated that LBP was strongly correlated with decreased LL, especially when affected patients were compared with age-matched healthy controls [22]. Additionally, Miyakoshi et al. reported that decreased LL and the limitation of total spinal extension are important risk factors for gait disturbance in patients with chronic LBP [23]. These findings indicate that osteoporosis patients with spinal sagittal malalignment, even those without VFs, had worse HRQoL and LBP compared with patients with normal spinal sagittal alignment. Further, spinal sagittal malalignment is a potential risk factor for LBP and HRQoL in patients with osteoporosis. 

The current study had some limitations. First, we did not evaluate medication status, such as use of painkillers and osteoporosis medications. In addition, we included patients with osteoporosis who first visited our department, although many patients had already undergone an intervention during their consultation. Painkillers and osteoporosis medication use might affect HRQoL as well as LBP. Second, this was a cross-sectional study; therefore, we could not evaluate cause-and-effect relationships among spinal sagittal malalignment, VFs, LBP, and HRQoL. Additionally, the patho-mechanism of spinal sagittal malalignment in patients without VFs remains unclear. To further understand these mechanisms, additional studies with larger sample sizes and a longitudinal design are needed.

## 5. Conclusions

The majority of patients with osteoporosis had spinal sagittal malalignment, and more than 70% of patients without VFs, had spinal sagittal malalignment. Furthermore, patients with spinal sagittal malalignment had worse LBP and HRQoL compared with patients with normal spinal sagittal alignment. These findings suggest that spinal sagittal malalignment is a potential risk factor for LBP and HRQoL in patients with osteoporosis.

## Figures and Tables

**Figure 1 jcm-10-02827-f001:**
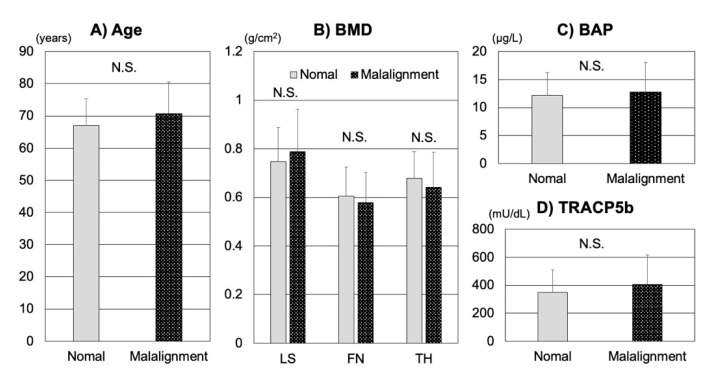
In patients without VFs, comparisons of (**A**) age, (**B**) BMD, (**C**) BAP, and (**D**) TRACP5b values determined in patients of the normal alignment and sagittal malalignment groups are shown. VF, vertebral fracture; N.S.: not significant; BMD, bone mineral density; BAP, bone-specific alkaline phosphatase; TRACP5b, tartrate-resistant acid phosphatase 5b.

**Figure 2 jcm-10-02827-f002:**
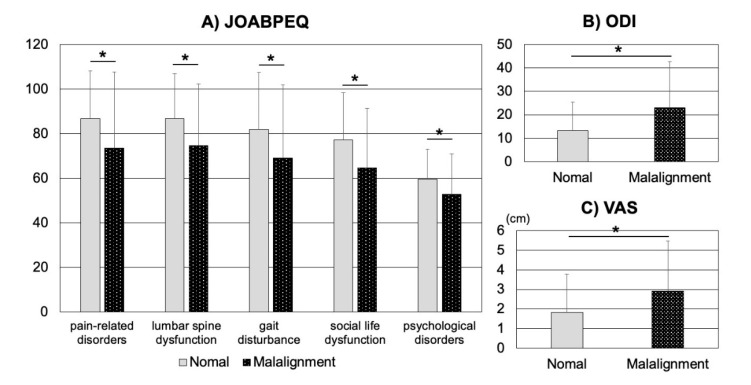
In patients without VFs, comparisons of (**A**) JOABPEQ, (**B**) ODI, and (**C**) VAS of LBP for patients of the normal alignment and sagittal malalignment groups are shown. VF, vertebral fracture; LBP, low back pain; ODI, Oswestry Disability Index; JOABPEQ, Japanese Orthopedic Association Back Pain Evaluation Questionnaire; VAS, Visual Analogue Scale, * *p* < 0.05.

**Figure 3 jcm-10-02827-f003:**
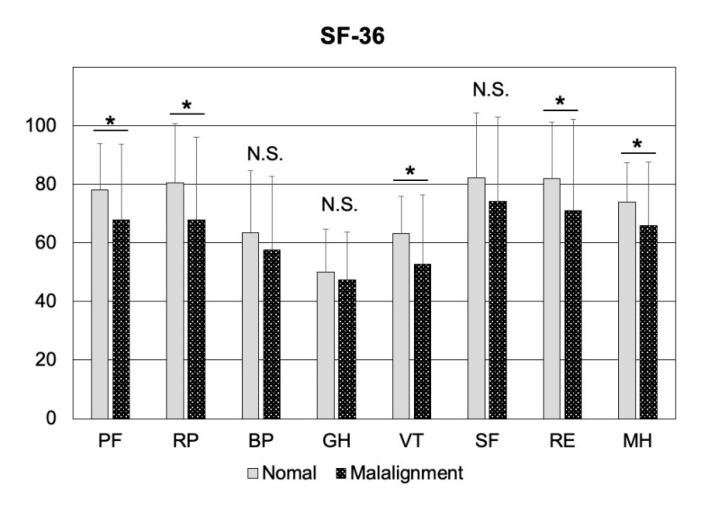
A comparison of MOS 36-Item Short-Form Health Survey scores of normal alignment and sagittal malalignment groups in patients without VFs is shown. VF, vertebral fracture; PF: Physical function, RP: Role physical, BP: Body pain, GH: General health, VT: Vitality, SF: Social functioning, RE: Role emotional, MH: Mental health, N.S.: not significant, * *p* < 0.05.

**Table 1 jcm-10-02827-t001:** The patient population of this study.

		Total		VF: 0		VF: 1		VF: ≥2		Comparison
N		259		129		50		80		-
Sex (M:W)		48:211		18:111		9:41		21:59		N.S.
		mean	SD	mean	SD	mean	SD	mean	SD	
Age		71.5	10.3	69.6	9.5	72.4	10.8	73.8	10.8	N.S.
BMD	LS	0.783	0.169	0.776	0.166	0.786	0.153	0.791	0.185	N.S.
	FN	0.558	0.118	0.586	0.122	0.543	0.099	0.522	0.113	VF: 0 vs. ≥2 *p* < 0.05
	TH	0.623	0.132	0.652	0.136	0.591	0.126	0.596	0.121	VF: 0 vs. 1, 0 vs. ≥2 *p* < 0.05
Bone turnover marker	BAP	14.8	12.5	14.2	14.6	15.3	10.0	15.5	9.9	N.S.
	TRACP5b	407	244	390	198	406	231	436	309	N.S.
Spinal sagittal alignment	PT	24.8	11.9	22.4	11.2	26.2	10.4	27.6	13.3	VF: 0 vs. ≥2 *p* < 0.05
	PI-LL	15.1	21.3	13.1	20.3	16.5	18.4	17.6	24.4	N.S.
	SVA	60.4	68.4	48.6	64.8	69.1	62.8	74.0	74.5	VF: 0 vs. ≥2 *p* < 0.05
JOABPEQ	pain-related disorders	77.3	31.6	77.3	31.6	62.2	32.4	66.1	32.2	VF: 0 vs. 1, 0 vs. ≥2 *p* < 0.05
	lumbar spine dysfunction	69.9	29.1	78.1	26.3	64.1	26.8	60.4	31.1	VF: 0 vs. 1, 0 vs. ≥2 *p* < 0.05
	gait disturbance	61.5	34.7	72.6	31.5	54.1	34.0	48.1	34.3	VF: 0 vs. 1, 0 vs. ≥2 *p* < 0.05
	social life dysfunction	58.2	27.5	68.2	25.8	52.4	26.7	45.5	24.5	VF: 0 vs. 1, 0 vs. ≥2 *p* < 0.05
	psychological disorders	49.6	17.3	54.7	17.3	44.2	15.9	44.9	15.8	VF: 0 vs. 1, 0 vs. ≥2 *p* < 0.05
ODI		26.7	20.7	20.3	18.5	29.3	19.6	35.5	21.3	VF: 0 vs. 1, 0 vs. ≥2 *p* < 0.05
VAS		3.3	3.8	2.6	2.4	3.7	3.0	4.2	5.5	VF: 0 vs. ≥2 *p* < 0.05
SF-36	PF	63.7	43.3	74.7	52.4	57.8	27.4	49.9	28.1	VF: 0 vs. ≥2 *p* < 0.05
	RP	60.3	31.7	71.5	26.8	54.0	32.1	46.3	32.2	VF: 0 vs. 1, 0 vs. ≥2 *p* < 0.05
	BP	53.4	25.4	59.3	24.1	46.8	25.1	47.9	25.6	VF: 0 vs. 1, 0 vs. ≥2 *p* < 0.05
	GH	44.9	16.8	48.2	15.8	42.4	18.0	41.2	16.9	VF: 0 vs. ≥2 *p* < 0.05
	VT	51.7	21.6	55.8	21.5	49.5	22.4	46.5	20.0	VF: 0 vs. ≥2 *p* < 0.05
	SF	69.7	28.7	76.4	27.3	64.0	26.3	62.4	29.9	VF: 0 vs. 1, 0 vs. ≥2 *p* < 0.05
	RE	63.8	33.4	74.1	28.8	56.1	32.0	51.9	36.2	VF: 0 vs. 1, 0 vs. ≥2 *p* < 0.05
	MH	63.7	20.8	68.3	20.0	59.7	21.7	58.8	20.1	VF: 0 vs. 1, 0 vs. ≥2 *p* < 0.05
		N	%	N	%	N	%	N	%	
normal alignment		54	20.8	36	27.9	7	14.0	11	13.8	*p* < 0.05
malalignment		205	79.2	93	72.1	43	86.0	69	86.3	

BMD: body mass index, LS: lumbar spine, FN: femoral neck, TH: total hip, BAP: bone-specific alkaline phosphatase, TRACP5b: tartrate-resistant acid phosphatase 5b, PT: pelvic tilt, PI-LL: pelvic incidence minus lumbar lordosis, SVA: sagittal vertical axis, VF: vertebral fracture, JOABPEQ: Japanese Orthopedic Association Back Pain Evaluation Questionnaire, ODI: Oswestry Disability Index, VAS: visual analogue scale of low back pain, SF-36: MOS 36-Item Short-Form Health Survey, PF: physical function, RP: role physical, BP: body pain, GH: general health, VT: vitality, SF: social functioning, RE: role emotional, MH: mental health.

## Data Availability

The data presented in this study are available on request from the corresponding author.

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
