# Peer review of "Prevalence and Characteristics of Spinal Sagittal Malalignment in Patients with Osteoporosis"

_jcm, 2021, doi:10.3390/jcm10132827_

Round 1
Reviewer 1 Report
The topic is of interest and is a potential field to explore in a prospective follow-up series.
The authors should precise wether the evaluated patients presented with sagittal imbalance before or after treatment of the osteoporotic fractures.
Author Response
We thank the reviewer for taking time to review our paper. As reviewer’s suggestion, we revised our manuscript as follows.
Reviewer #1
- The topic is of interest and is a potential field to explore in a prospective follow-up series. The authors should precise whether the evaluated patients presented with sagittal imbalance before or after treatment of the osteoporotic fractures.
Response to comment from Reviewer #1-1
We thank the reviewer for this suggestion. The current study was a cross-sectional study, and patients were included at their first visit to our department. However, many patients had already undergone an intervention during their consultations. Furthermore, we excluded patients with new vertebral fractures, and only patients who could achieve bone union were included in this study. Therefore, we could not evaluate the impact of the treatment on osteoporosis and fractures. We revised and added these sentences to the Materials and Methods section and the Discussion section as follows:
The records of patients with osteoporosis who first visited these facilities from June 2015 to March 2017 were reviewed in this cross-sectional study. We excluded patients who developed new vertebral fractures within 3 months. The remaining 259 patients (48 men, 211 women; mean age: 71.5 years) were included.
In addition, we included patients with osteoporosis who first visited our department, although many patients had already undergone an intervention during their consultation.
Reviewer 2 Report
The authors studied 259 osteoporotic patient and found that 79% had sagittal imbalance by Schwab criteria due to compression fractures. This paper highlights the high incidence of sagittal imbalance in an increasingly older population and the impaired quality of life due to fixed deformity. The methods are sound and the data are useful. A thorough analysis of bone health was performed across the balanced and imbalanced groups. A valuable contribution to the literature
Author Response
We thank the reviewer for taking time to review our paper. As reviewer’s suggestion, we revised our manuscript as follows.
Reviewer #2
The authors studied 259 osteoporotic patient and found that 79% had sagittal imbalance by Schwab criteria due to compression fractures. This paper highlights the high incidence of sagittal imbalance in an increasingly older population and the impaired quality of life due to fixed deformity. The methods are sound and the data are useful. A thorough analysis of bone health was performed across the balanced and imbalanced groups. A valuable contribution to the literature
Response to comment from Reviewer #2
We sincerely appreciate your great effort in reviewing the manuscript. Your comments have encouraged our current and future studies.
Reviewer 3 Report
Introduction:
Line 50 delete: adult
Materials and Methods:
Line 80 better explains the LL measurement
Results:
You could add data on pain and quality of life in patients with fractures
Discussion
The discussion is fragmentary and disconnected.
Recommends to review and improve it
Eexample: line 71 the statement is not supported by the results
I advise you to consult these two articles would be for you:
1)Effectiveness of rehabilitative intervention on pain, postural balance, and quality of life in women with multiple vertebral fragility fractures: A prospective cohort study. Scaturro, D., Rizzo, S., Sanfilippo, V., ...Iolascon, G., Mauro, G.L.Journal of Functional Morphology and Kinesiology, 2021, 6(1), 24
2)Is there a relationship between mild-moderate back pain and fragility fractures? Original investigation. Scaturro, D., Lauricella, L., Tumminelli, L.G., Tomasello, S., Mauro, G.L.Acta Medica Mediterranea, 2020, 36(3), pp. 2149–2153
Author Response
We thank the reviewer for taking time to review our paper. As reviewer’s suggestion, we revised our manuscript as follows.
Reviewer #3
- Introduction: Line 50 delete: adult
Response to comment from Reviewer #3-1
We removed the word in agreement with the reviewer.
- Materials and Methods: Line 80 better explains the LL measurement
Response to comment from Reviewer #3-2
We revised this text according to the reviewer’s suggestion as follows,
To measure lumbar lordosis (LL), we assessed the angles between the first line parallel to the upper endplate of L1 and the second line parallel to the superior endplate of the sacral base on lateral views of the whole-spine radiograph.
- Results: You could add data on pain and quality of life in patients with fractures
Response to comment from Reviewer #3-3
We thank the reviewer for this suggestion. We added data on LBP and SF-36 scores for patients with fractures (Table 1). Further, the results of the comparisons were added as follows:
With regard to LBP and the HRQoL score, all five domains of JOABPEQ, ODI, and VAS as well as all eight subscales of the SF-36 of the ≥2 VF group were significantly worse than those of the 0 VF group (p < 0.05). In addition, the values of all five domains of JOABPEQ, ODI, RP, BP, SF, RE, and MH of SF-36 in the 1 VF group were significantly worse than those in the 0 VF group (p < 0.05) (Table 1).
Furthermore, as the number of VFs increased, the prevalence of spinal sagittal malalignment also increased and LBP and HRQoL scores worsened.
- Discussion: The discussion is fragmentary and disconnected. Recommends to review and improve it. Example: line 71 the statement is not supported by the results. I advise you to consult these two articles would be for you:
1) Effectiveness of rehabilitative intervention on pain, postural balance, and quality of life in women with multiple vertebral fragility fractures: A prospective cohort study. Scaturro, D., Rizzo, S., Sanfilippo, V., ...Iolascon, G., Mauro, G.L. Journal of Functional Morphology and Kinesiology, 2021, 6(1), 24
2) Is there a relationship between mild-moderate back pain and fragility fractures? Original investigation. Scaturro, D., Lauricella, L., Tumminelli, L.G., Tomasello, S., Mauro, G.L. Acta Medica Mediterranea, 2020, 36(3), pp. 2149–2153
Response to comment from Reviewer #3-4
We thank the reviewer immensely for providing this suggestion and these great reference articles, which have great potential in improving our manuscript. We consulted these reference articles and reviewed our discussion section for improvements. Further, we added new text to the Discussion section and new references as per the reviewer’s suggestion as follows:
Regarding the relationship between VFs and spinal sagittal alignment, as the number of VFs increased, the prevalence of spinal sagittal malalignment also increased, and LBP and HRQoL scores worsened in this study. Mochizuki et al. previously reported that spinal sagittal alignment is associated with age and VF in patients with rheumatoid arthritis [10]. In addition, osteoporosis patients with VFs have worse global sagittal alignment and a worsened quality of life [6]. Scaturro et al. reported that the severity of LBP is correlated with the number of vertebral fractures [@]. These findings indicate that VFs are closely correlated with sagittal spinal malalignment and affect LBP as well as HRQoL.
With regard to cause-and-effect relationships between VFs and spinal sagittal malalignment, Zhang et al. reported that multiple VFs lead to spinal sagittal malalignment in patients with osteoporosis [11]. In contrast, several authors reported that spinal sagittal malalignment was a potential risk factor for increased VF incidence in patients with osteoporosis [4, 12, 13]. These findings indicate that VFs induce spinal sagittal malalignment; spinal sagittal malalignment also leads to VFs in patients with osteoporosis.
In the current study, more than 70% of patients with osteoporosis without VFs had spinal sagittal malalignment. In a longitudinal study with a minimum of 10 years of follow-up, Takeda et al. reported that spinal sagittal malalignment, decreased lumbar lordosis, and increased SVA were correlated with age in patients without VFs [14]. Regarding the underlying mechanism of spinal sagittal malalignment in patients without VFs, several authors reported a relationship between spinal sagittal malalignment and decreased muscle mass in patients with spinal diseases [15, 16]. Additionally, Scaturro et al. reported that combination treatments with medication and postural training/resistance exercises showed improvements in the pain and QoL for patients with osteoporosis undergoing rehabilitation [@]. These findings indicate that decreasing muscle mass may induce spinal sagittal malalignment.
Is there a relationship between mild-moderate back pain and fragility fractures? Original investigation. Scaturro, D., Lauricella, L., Tumminelli, L.G., Tomasello, S., Mauro, G.L. Acta Medica Mediterranea, 2020, 36(3), pp. 2149–2153
Effectiveness of rehabilitative intervention on pain, postural balance, and quality of life in women with multiple vertebral fragility fractures: A prospective cohort study. Scaturro, D., Rizzo, S., Sanfilippo, V., ...Iolascon, G., Mauro, G.L. Journal of Functional Morphology and Kinesiology, 2021, 6(1), 24
Round 2
Reviewer 3 Report
All required changes have been made by improving the manuscript